# Metabolomics of sebum reveals lipid dysregulation in Parkinson's disease

Eleanor Sinclair[1], Drupad K. Trivedi[1], Depanjan Sarkar[1], Caitlin Walton-Doyle[1], Joy Milne[1], Tilo Kunath [2], Anouk M. Rijs [3], Rob M. A. de Bie [4], Royston Goodacre [5], Monty Silverdale[6] & Perdita Barran [1✉]

Parkinson's disease (PD) is a progressive neurodegenerative disorder, which is characterised by degeneration of distinct neuronal populations, including dopaminergic neurons of the substantia nigra. Here, we use a metabolomics profiling approach to identify changes to lipids in PD observed in sebum, a non-invasively available biofluid. We used liquid chromatography-mass spectrometry (LC-MS) to analyse 274 samples from participants (80 drug naïve PD, 138 medicated PD and 56 well matched control subjects) and detected metabolites that could predict PD phenotype. Pathway enrichment analysis shows alterations in lipid metabolism related to the carnitine shuttle, sphingolipid metabolism, arachidonic acid metabolism and fatty acid biosynthesis. This study shows sebum can be used to identify potential biomarkers for PD.

[1] Manchester Institute of Biotechnology, School of Chemistry, The University of Manchester, Manchester, UK. [2] Institute for Stem Cell Research, School of Biological Sciences, The University of Edinburgh, Edinburgh, UK. [3] Division of BioAnalytical Chemistry, AIMMS Amsterdam Institute of Molecular and Life Sciences, Vrije Universiteit Amsterdam, Amsterdam, The Netherlands. [4] Department of Neurology, Amsterdam Neuroscience, Amsterdam University Medical Centers, University of Amsterdam, Amsterdam, The Netherlands. [5] Institute of Systems, Molecular and Integrative Biology, Department of Biochemistry and Systems Biology, University of Liverpool, Liverpool, UK. [6] Department of Neurology, Salford Royal Foundation Trust, Manchester Academic Health Science Centre, University of Manchester, Manchester, UK. ✉email: perdita.barran@manchester.ac.uk

Parkinson's disease (PD) is a neurodegenerative disorder affecting over 6 million globally, second only in prevalence to Alzheimer's disease[1]. The principal pathological hallmark of PD is the formation of aggregated α-synuclein deposits in the brainstem, which are the major components of Lewy bodies[2,3]. The disease is also characterised by the loss of dopaminergic neurons in the substantia nigra pars compacta producing a decline in striatal dopamine levels and subsequent loss of motor function[4]. There is no conclusive preclinical diagnostic test for PD. Clinical diagnosis is achieved primarily through observations by a physician, of the decline in motor functions[5,6]. These clinical manifestations normally present as a combination of one or more of the four cardinal signs of PD, namely; bradykinesia, resting tremor, rigidity, and postural instability[7,8]. A formal diagnosis often occurs following the depletion of 60–80% of the brains dopaminergic neurons[2]. Non-motor symptoms are thought to precede motor symptoms by upto 20 years, some of these include: mood disorders, sleep disorders, and olfactory deficits[9,10]. Seborrhoeic dermatitis is a common non-motor symptom reported in up to 60% of people with Parkinson's (PwP)[11,12]. This condition presents as "oily skin" that correlates to an excess of sebum, produced and secreted by the sebaceous glands in the dermis of the skin. Sebum is a complex lipid-rich substance that is predominantly composed of triglycerides, fatty acids, wax esters, squalene, and cholesterol[13]. It serves as a protective agent to the skin providing waterproofing, thermoregulation, and photoprotection, alongside suggested antimicrobial and antioxidant activities[14,15]. Studies of sebum are commonplace in dermatological conditions such as acne, however sebum as a biofluid has rarely been used in disease diagnostics. In our recent study, we have reported the presence and differential regulation of volatile organic compounds in the sebum of PwP[16].

The analysis of complex mixtures of metabolites present in a lipid-rich biofluid such as sebum, calls for a sensitive and robust analytical platform. Mass spectrometry (MS) is a leading analytical technique for clinical metabolomics analyses and when hyphenated to chromatography, benefits from increased resolution and sensitivity[17,18]. Liquid chromatography-mass spectrometry (LC-MS) facilitates the qualitative and quantitative analysis of the wide range of molecular species found within complex mixtures such as sebum. LC-MS has been used to study a number of biofluids in relation to PD prognosis and diagnosis, such as blood, saliva, and cerebrospinal fluid (CSF)[19–25]. Alterations in the expression of metabolites and the downstream effects on their corresponding metabolic pathways have also been extensively studied for PD diagnostics within the blood and CSF metabolome, including: catecholamines, dopamine metabolites, amino acids, and urate alongside fatty acid metabolism, energy metabolism, and kynurenine metabolism[19,22,26–29]. The use of sebum as a diagnostic tool for PD provides an exciting prospect from which a non-invasive and inexpensive test could be developed to detect the onset of the disease. In this study, we have used LC-MS to separate and detect lipid-like species and small molecules present in sebum. We have used data-driven approaches, with robust statistical validation, to discover biomarkers of Parkinson's disease present in sebum. This will inform the development of future PD biomarkers alongside the understanding of metabolic pathways altered in PD. Additionally, we also investigate whether variations in the measured sebum metabolome between early drug naïve PD and later medicated PD were observed, suggesting changes in the metabolic pathways during disease progression.

## Results

**Analysis of patient metadata.** The study population comprised of 274 participants which included 138 medicated PD, 80 drug

**Table 1 Demographics of participants included in classification modelling and statistical analysis.**

| Parameters | Independent control | Drug naïve PD | Medicated PD |
|---|---|---|---|
| $n$ | 56 | 80 | 138 |
| Age (years)[a] | 54.3 ± 14.4 | 69.8 ± 9.4 | 70.3 ± 8.2 |
| BMI (kg/m²)[a] | 26.1 ± 4.4 | 25.8 ± 4.9 | 26.3 ± 5.4 |
| Gender (Male:Female)[b] | 0.87 | 1.76 | 1.65 |
| Alcohol intake (Yes:No)[c] | 4.60 | 1.76 | 1.81 |
| Smoker (Yes:No)[c] | 0.08 | 0.07 | 0.00 |

BMI body mass index.
[a]BMI and age values are expressed as mean ± standard deviation.
[b]Expressed as a ratio (Male:Female).
[c]Expressed as a ratio (Yes:No).

naïve PD and 56 control subjects. An overview of important patient demographics is summarised in Table 1. The results of significance tests between cohort group metadata are reported in Supplementary Table 1. Two-tailed Mann–Whitney U-test showed age is significantly different ($p < 0.05$) between control and PD cohorts (both drug naïve and medicated), however, BMI was not statistically significantly between these groups. There were more male participants in both PD cohorts (M/F > 1.5) compared to a higher proportion of female participants within the control group (M/F < 1). This was perhaps expected as the higher incidence and prevalence rates of PD in the male population is recognised and studies show a 1.4–1.5 fold increase in the number of male PD cases, although the reason for this is not yet understood[1,30]. A similar comparison of the number of participants who smoke (yes/no) or consume alcohol (yes/no) showed no significant differences between drug naïve PD and control cohorts, with p-values of 0.837 and 0.192, respectively. However, the number of participants who consume alcohol was found to be 2.5 times higher in the control group compared to medicated PD. There were no smokers in the medicated PD cohort and 7% within the control group, which was deemed significant by a Fisher's exact test (p-value 0.006). The discovery of significant differences of these metadata parameters between PD and control cohorts has led us to test their impact on classification accuracy, which are described within the following results sections.

**Data driven prediction of PD.** In order to assess variation between the measured metabolome by phenotype, partial least squares-discriminant analysis (PLS-DA) was used. Two PLS-DA models were constructed, each using a two-class input: (1) drug naïve PD vs. control and (2) medicated PD vs. control. It is well known that unbalanced numbers within classification groups may bias prediction accuracy towards the majority class and to overcome this here, Synthetic Minority Over Sampling Technique (SMOTE) was applied[31]. PLS-DA models were built and validated using bootstrap resampling with replacement ($n = 250$). Figure 1 reports the classification sensitivity and specificity rates of each PLS-DA model alongside the observed and null distributions (from permutation testing).

To evaluate if gender influenced classification accuracy, two PLS-DA models were built for each gender separately, for drug naïve PD vs. control and medicated PD vs. control. If the compounds accounting for variance between disease and control were gender specific, we could expect consistent and significantly higher sensitivity and specificity values for one gender, which we did not find to be true (see Supplementary Table 3). Combined

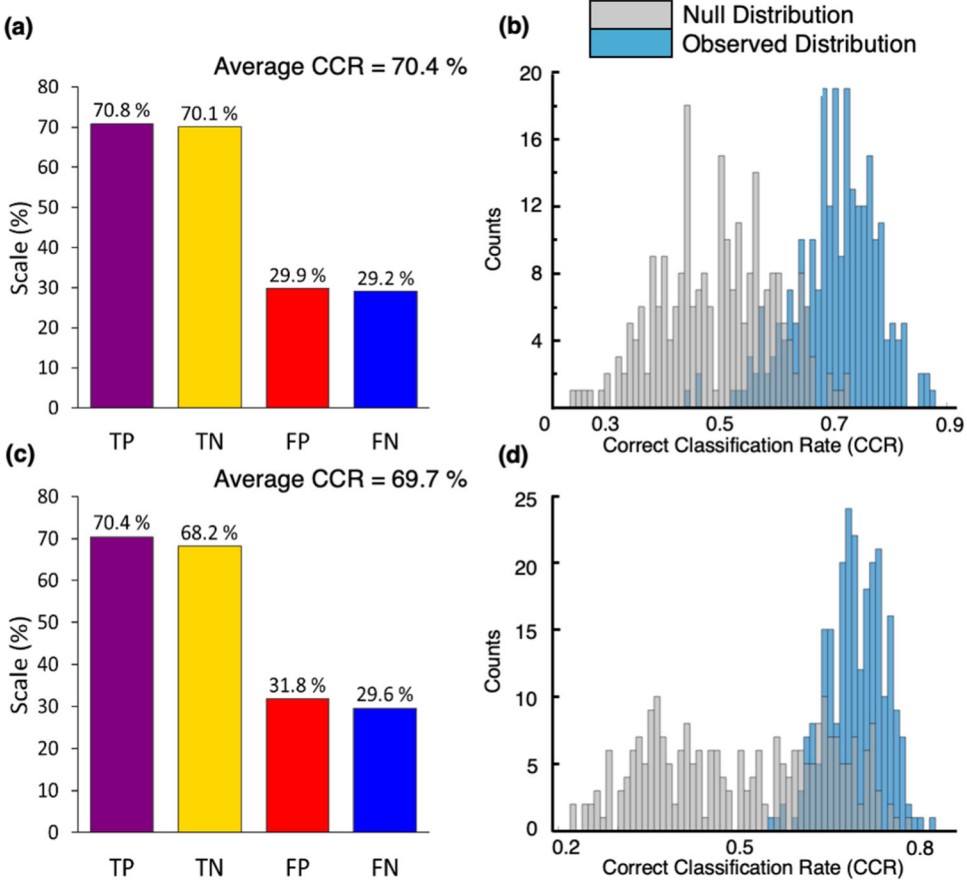

**Fig. 1 PLS-DA classification models for (a, b) drug naïve PD vs. control and (c, d) medicated PD vs. control. a, c** Classification rates for each model including true positive (TP, sensitivity), true negative (TN, specificity), false positive (FP), and false negative (FN). **b, d** Null distribution (grey bars) and observed distribution (blue bars) for each PLS-DA bootstrap model. The correct classification rate (CCR) were calculated from the test sets only (n = 250 from the bootstraps).

gender models (Fig. 1) were used for subsequent analysis owing to the heightened power attributed to statistical models with larger input groups. PLS-DA was also used to determine if geographical location or variances between clinician sampling could impact classification using an independent control cohort. Samples (n = 40) were chosen from four recruitment clinics, located in the north (n = 2) and south (n = 2) of the UK. Confounding factors were controlled so that age and BMI were not statistically significant between groups (one-way ANOVA p-value > 0.05) and the male-to-female ratio was identical. The average CCR for this model was 21%, which therefore indicates that our data is not biased by recruitment site or the clinician who collected samples.

**Selection of significant features which classify PD.** To define the features responsible for the measured variance in PLS-DA prediction models, variable importance in projection (VIP) scores were calculated. Receiver operating characteristic (ROC) analysis was performed on variables with VIP score > 1 (Fig. 2). The number of variables that met this threshold were 15 in Drug naïve PD and 26 in medicated PD analyses. The area under the curve (AUC) and 95% confidence intervals (CI) for each individual variable obtained from univariate ROC curve analysis are reported in the Supplementary Fig. 2. A limitation in ROC analysis of individual features is the failure to consider relationships between the features that account for the observed variance. The outcome of a multivariate analysis is reduced to a univariate one, in which each individual feature is treated as the sole biomarker accounting for 100% of the variation between the classes.

Therefore, in combination with assessing individual metabolite ROC curves, a multivariate ROC analysis approach was also implemented based on the PLS-DA method (Fig. 2a, b).

We note that PLS-DA could not accurately differentiate medicated PD and drug naïve PD. Sensitivity and specificity values of 59.7 and 50.3% were returned for PLS-DA models in which medicated PD was the "positive" predicting class (data shown in Supplementary Fig. 1). Figures 2a, 2b report ROC curves for drug naïve PD and medicated PD models, respectively, which each use all VIP compounds > 1 for each respective model. VIP score examination of drug naïve PD vs. control and medicated PD vs. control models confirms that ten variables (VIP > 1) are common between the two PD groups. To investigate biomarkers associated with the diagnosis of PD rather than disease stage stratification and to avoid possible effect of medication, the common metabolites (VIP > 1) between drug naïve and medicated PD analyses were evaluated further. Figure 2c presents a multivariate ROC analysis for each common variable, and this analysis reports increased sensitivity and specificity rates as a function of the number of variables included in each model as demonstrated by higher AUC values. In addition, the 95% confidence interval range decreases as the number of variables in each model increases.

Pearson correlation coefficients were calculated for each significant variable (VIP > 1) to investigate association of alcohol and significant variables. None of the significant compounds are associated to an increase in alcohol consumption (Supplementary Fig. 3). To exclude the possible contribution of age to disease classification, age was

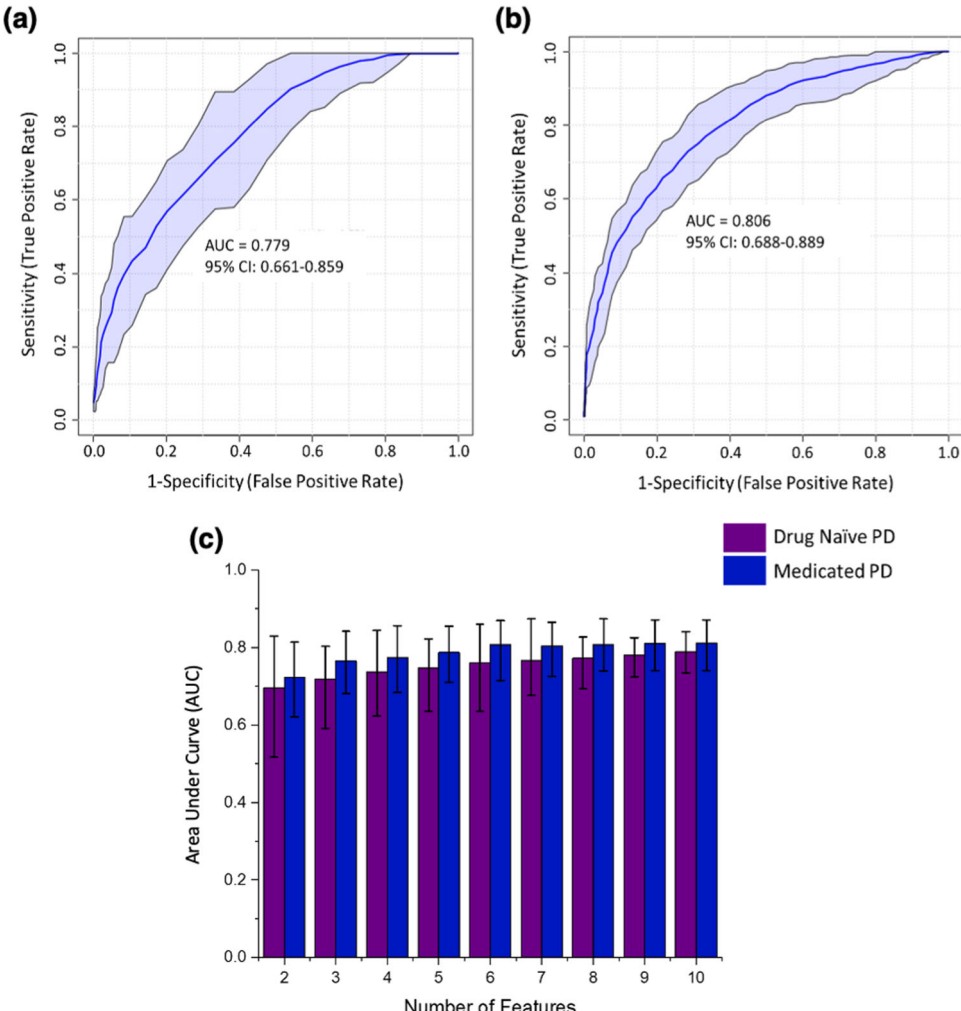

**Fig. 2 ROC curve analyses based on a multivariate PLS-DA algorithm with a two latent variable input, AUC and 95% confidence intervals (CI) were calculated by Monte Carlo cross validation (MCCV) using balanced subsampling with multiple repeats. a** ROC curve analysis ($n = 15$ independent metabolite features) in drug naïve PD vs. control PLS-DA with VIP > 1. **b** ROC curve analysis ($n = 26$ independent metabolite features) in medicated PD vs. control PLS-DA with VIP > 1. **c** A bar chart displaying the comparison of AUCs for drug naïve PD (purple) and medicated PD (blue) using common VIPs between models ($n = 10$ independent metabolite features), data are presented as mean AUC value with error bars representing the minima and maxima values of the 95% CI range.

included as a variable for further PLS-DA models, giving it equal weighting as any other measured variable. If age had any significance, it had equal chances to contribute to the model and would be ranked as high as other measured variables. The difference in CCR between models with and without the inclusion of age were negligible (<0.5%), and VIP scores for the age variable were $1.17 \times 10^{-11}$ and $2.11 \times 10^{-11}$ for drug naïve and medicated PD models, respectively. In perspective, the variables were ranked at 6492 and 6498 out of a possible 6502 ranks, which strongly indicates that age is not a contributing factor for the separation presented.

**Annotation of metabolites associated with PD diagnosis.** Metabolomics Standards Initiative (MSI) guidelines[32] and International Lipid Classification and Nomenclature Committee (ILCNC)[33] guidelines were adhered to for the annotation of common significant metabolites ($n = 10$) (Table 2). Table 2A reports putative annotations based upon accurate mass and tandem MS fragmentation data for five of the significant compounds (MSI level 2). Table 2B reports the database matches based upon accurate mass, although there are no fragmentation data to support these matches the only possible hits from two databases (Lipid Maps and

METLIN) within a low mass tolerance (10 ppm) correspond to a single chemical formula in three of the five compounds; the remaining two compound had no matches. Ceramides, triacylglycerol, glycosphingolipid, and fatty acyl lipid classes were amongst those putatively annotated in both common and non-common VIP compounds. Putative annotations and database matches listed in Table 2A, B are expounded upon in Tables S3A and S3B, respectively. Notably, metabolites belonging to ceramide, triacylglycerol, and fatty acyl classes were downregulated whereas glycosphingolipid and fatty acyl metabolites were upregulated in PD. Box plots comparing control, drug naïve PD and medicated PD cohorts for these compounds are displayed in Fig. 3. Further details of putative compound annotations for all metabolites with VIP score > 1 in drug naïve PD and medicated PD analyses are found in Supplementary Tables 4A, B and 5A, B, respectively.

**Sebum metabolome measurements: context to current understanding of PD.** Pathway enrichment analysis was performed to explore changes in metabolic pathways with respect to disease onset and progression. A prerequisite for traditional pathway analysis methods is the annotation of all analytically detected

**Table 2 Putative annotations of the ten VIP compounds common between drug naïve PD and medicated PD analyses (VIP > 1).**

**(A) Putative annotations have been assigned using accurate mass and MS/MS fragmentation matched against Lipid Maps database (LMSD) and Lipid Blast.**

| Feature | Putative annotation (Accurate mass & MS/MS fragmentation) | Expression drug naïve PD (fold change) | Expression medicated PD (fold change) |
|---|---|---|---|
| *m/z* 825.6939 | TG(50:5) | ↓ (0.77) | ↓ (0.64) |
| *m/z* 764.5681 | HexCer(36:2) | ↑ (1.15) | ↑ (1.10) |
| *m/z* 666.6370 | Cer(42:0) | ↓ (0.60) | ↓ (0.47) |
| *m/z* 638.6067 | Cer(40:0) | ↓ (0.61) | ↓ (0.47) |
| *m/z* 610.5763 | Cer(38:1) | ↓ (0.63) | ↓ (0.48) |

**(B) Putative annotations have been assigned using accurate mass measurements matched against Lipid Maps (LMSD and COMP_DB) and METLIN databases.**

| Measured feature | Database match(s) (accurate mass) | Formula | Expression drug naïve PD (Fold change) | Expression medicated PD (Fold change) |
|---|---|---|---|---|
| *m/z* 414.4308 | FA(26:0) Methyl pentacosanoate | $C_{26}H_{52}O_2$ | ↑ (1.23) | ↓ (0.84) |
| *m/z* 358.3677 | FA(22:0)* | $C_{22}H_{44}O_2$ | ↓ (0.81) | ↓ (0.78) |
| *m/z* 194.1396 | FA(8:0) ʟ-Cladinose Metaldehyde† | $C_8H_{16}O_4$ | ↑ (1.74) | ↑ (1.78) |
| *m/z* 550.6277 | – | – | ↑ (1.33) | ↑ (1.10) |
| *m/z* 368.4242 | – | – | ↓ (0.15) | ↓ (0.14) |

*TG* triacylglyceride, *HexCer* hexosylceramide, *Cer* ceramide *FA* fatty acyl.
†Pesticide.

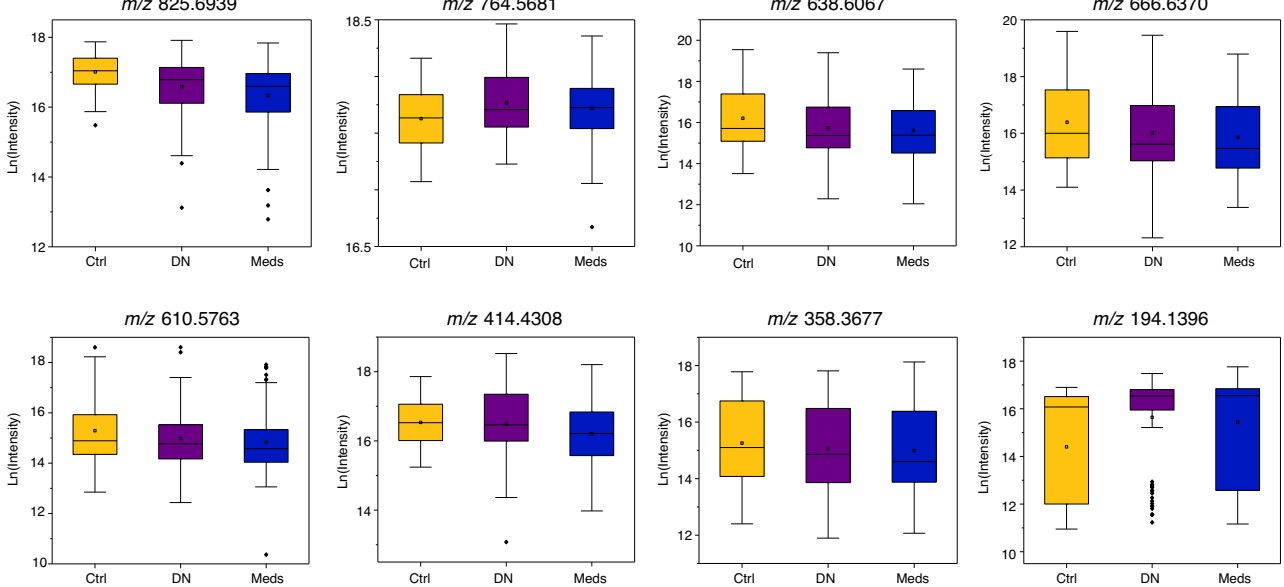

**Fig. 3 Box whisker plots for each of the eight putatively annotated compounds for control (Ctrl, yellow) (*n* = 56 biologically independent samples), drug naïve PD (DN, purple) (*n* = 80 biologically independent samples) and medicated PD (Meds, blue) cohorts (*n* = 138 biologically independent samples).** Box plots display mean (square), median (line within box) and quartiles (box limits), range (whiskers) and outliers (diamond). The *y*-axis of each plot corresponds to the natural log of intensity values and the measured *m/z* value for each compound is labelled above the plot, these species correlate to the data presented in Table 2A, B.

features via spectral and compound database matching. This is a major bottleneck in untargeted metabolomics workflows and due to the large number of features detected in this study, Mummichog analysis was employed[34]. The analysis was performed independently for the two PD cohorts using a Student's *t*-test (*p*-value < 0.05) between control subjects vs. (1) drug naïve PD and (2) medicated PD. There were 1378 and 504 features for drug naïve PD and medicated PD, respectively, which were significant between disease and control groups. Further details of significantly enriched pathways associated with PD can be found in Supplementary Tables 6 and 7 for drug naïve PD and medicated PD, respectively.

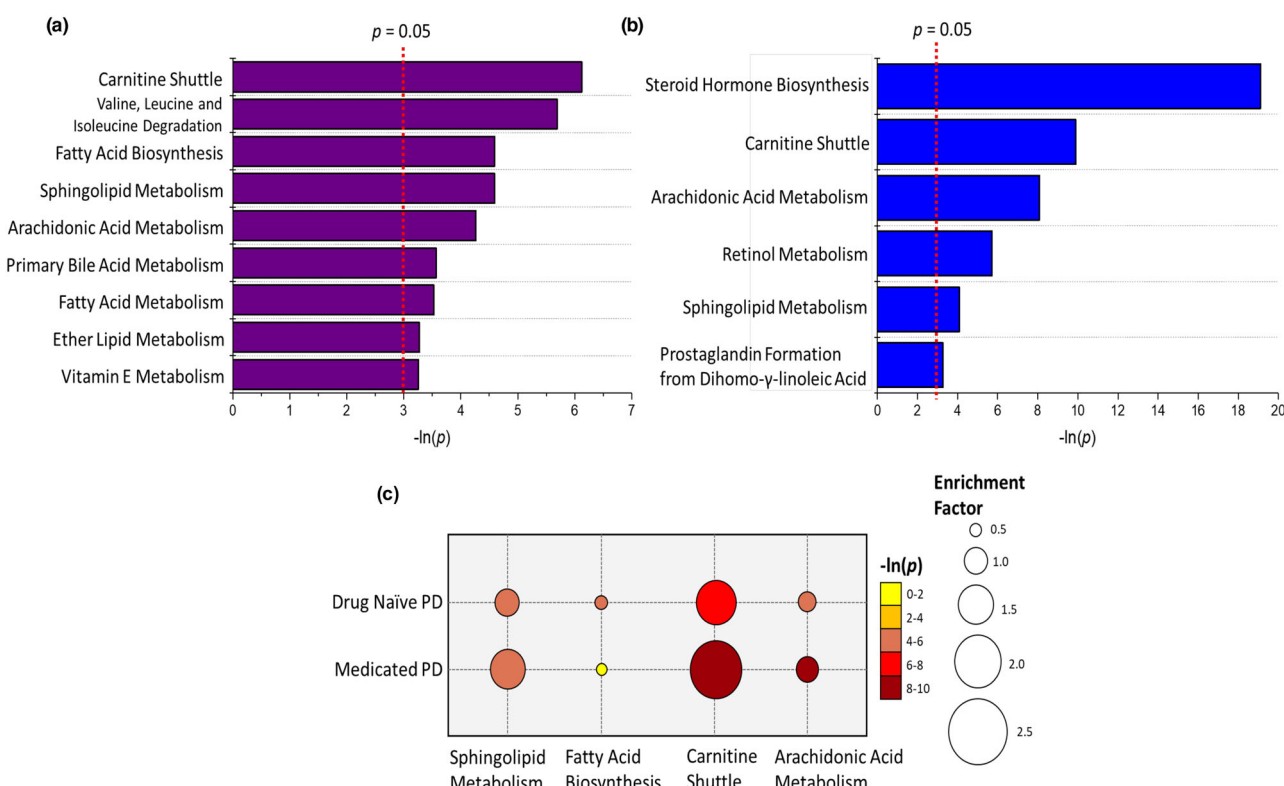

**Fig. 4 Results of mummichog analysis for significant pathways (p < 0.05).** Bar charts report pathways for (**a**) drug naïve PD vs. control and (**b**) medicated PD vs. control. **c** A bubble chart displaying the common significant pathways between drug naïve PD and medicated PD compared against controls; the bubble size refers to the enrichment factor of the pathway and the colour represents the natural log of the pathway p-value.

Mummichog analysis reveals the carnitine shuttle to be the most important pathway linked to drug naïve PD patients ($p = 0.002$) (Fig. 4a). This pathway increases in significance ($p = 5.09 \times 10^{-5}$) and enrichment within the medicated PD cohort, this can be visualised in Fig. 4c. The carnitine shuttle is highly involved in energy metabolism through the facilitation of long chain fatty acid (LCFA) β-oxidation via assisted transportation into the mitochondria by acyl-carnitine substrates[35]. Decreased long-chain acyl-carnitines, associated with insufficient β-oxidation, have previously been reported as potential diagnostic markers for PD[28]. The dysregulation of carnitine shuttle and vitamin E pathways have also been observed in frail elderly cohorts (between 56 and 84 years old) compared to resilient age-matched individuals[36]. The mapped $m/z$ features correspond to a series of differing length fatty acid chains of acyl-carnitine conjugates. As the carnitine shuttle is a mediation pathway for fatty acid oxidation, it is reasonable that the perturbation of fatty acid biosynthesis and fatty acid metabolism pathways could be linked, which is further supported by the putative assignment of associated compound classes to VIP compounds.

Additional compounds putatively annotated from PLS-DA models (VIP > 1) belong to the sphingolipid class of compounds (Table 2). The sphingolipid metabolism pathway was enriched in both drug naïve and medicated PD. Sphingolipids are a major lipid class that are abundant in lipid-rich structures of the body (such as skin) and have central roles in cell signalling and regulation. Interestingly, disruption to the sphingolipid metabolism has been reported as a downstream effect of increased α-synuclein levels[37,38] and α-synuclein is disrupted in PD skin[39]. Perturbations within the sphingolipid pathway have been previously linked to defects in both lysosomal and mitochondrial metabolism, which are often implicated in the pathogenesis of neurodegenerative diseases such as PD and Gaucher's disease[38,40–43]. Interestingly, the link between mitochondrial dysregulation and PD has been widely established in skin fibroblasts, however, never before in sebum[44,45]. Recent studies have found the dysregulation of ceramide levels, which are common structural units of all sphingolipids, in numerous diseases including PD, Alzheimer's disease and depression, although the general consensus from studies of sphingolipids in PD is an increase in ceramide levels[46–48]. Due to their bioactive role within cell membranes sphingolipids are strongly linked to sterol metabolism pathways, and have an established role in the modulation of steroidogenesis. There is a direct link between ceramides and the biosynthesis of cholesterol which is then the feed in substrate for steroid hormone biosynthesis, the most significantly altered pathway shown for medicated PD patients[49,50].

In conclusion, an untargeted LC-MS analysis of sebum obtained non-invasively from a simple skin swab from people with Parkinson's reveals a difference in the composition of sebum compared to control subjects. The overlap of ten metabolites from separate statistical analyses for drug naïve PD and medicated PD strengthens the evidence, that these compounds are associated with PD and not associated with dopaminergic medication. This is further supported by the identification of common pathways between the two PD classes that are significantly enriched. Insufficient clinical data is available for these patients to hypothesise on the ability of a sebum analysis to help stratify disease progression, although it should be included in further studies. Future work will also focus on targeting the putatively identified lipid classes, with the inclusion of ion mobility to enhance separation and increase the confidence in compound identification.

## Methods

**Sample participants**. The participants included in this study were part of a nationwide recruitment process taking place at 25 different NHS clinics, in addition to subjects ($n = 4$) that participated in a clinical trial in the Netherlands[51]. A total of 274 participants were recruited from three subject groups: control ($n = 56$), drug naïve PD ($n = 80$), and medicated PD ($n = 138$). The participants included in this study were selected at random from these sites. Ethical approval for this project (IRAS project ID 191917) was obtained by the NHS Health Research Authority (REC reference: 15/SW/0354). Informed consent was received from all participants prior to their enrolment in the study.

**Chemicals and materials**. The chemicals and materials utilised in this study were: gauze swabs (Arco, UK), sample bags (GE Healthcare Whatman$^{TM}$, UK), 15 mL and 50 mL centrifuge tubes (Greiner Bio-One, UK), microcentrifuge tubes 2 mL (Eppendorf, UK), Ministart® 0.2 μm syringe filter (Sartorius, UK), Optima® LC-MS grade solvents 2-propanol, acetonitrile, methanol, and formic acid (Fisher Scientific), HPLC grade HiPerSolv CHROMANORM® ethanol absolute (99.8%), CHROMASOLV$^{TM}$ LC-MS grade water (Honeywell) and Leucine Enkephalin (Waters, Wilmslow, UK).

**Sample collection**. Using a standard sampling procedure, each participant was swabbed by a clinician on the upper back with cotton-based medical gauze (7.5 cm × 7.5 cm) to collect sebum present on the skin. The sampled gauze swabs were sealed in background-inert plastic bags and transported to the central facility at the University of Manchester, where they were stored at −80 °C until end of recruitment.

**Sample extraction**. Gauze swabs were removed from −80 °C storage and allowed to equilibrate to room temperature. A solvent extraction method was used to prepare the samples for LC-MS analysis. Each gauze swab was transferred to an inert glass bottle. Methanol (9 mL) was added to each glass bottle and followed by vortex-mixing (10 s) and sonication (30 min) at ambient temperature, to extract sebum metabolites from gauze. The extracted metabolite-rich methanol was decanted from the gauze swab bottle and this solution was passed through a filter (0.2 μm). A recovery rate of approximately 7 mL per sample was achieved, which was aliquoted into three 2 mL fractions and one 1 mL fraction. Each 2 mL fraction was vacuum concentrated (Eppendorf) at ambient temperature for 12 h to remove methanol, which resulted in three identical sebum extracts per patient sample. These dried pellets were stored at −80 °C until required for analysis. A portion of the remaining 1 mL liquid fraction of each sample (100 μL) was used to create a biological pooled quality control (QC) sample. The mixture was vacuum centrifuged (Eppendorf) for 12 h at ambient temperature and the dried extract stored at −80 °C until analysis.

**Sample reconstitution**. Prior to LC-MS analyses dried sebum extracts were equilibrated to ambient temperature before reconstitution. Extracts were resuspended in 200 μL of MeOH:EtOH (v/v, 50:50). Samples were vortex-mixed (20 s), sonicated (5 min), and centrifuged (Eppendorf) at 12,000 × g for 10 min. The recovered supernatant (160 μL) was then submitted for LC-MS analysis.

**LC-MS analysis**. LC-MS analysis was performed on an Ultimate 3000 UHPLC (Thermo Scientific) coupled to a Synapt G2-Si QToF mass spectrometer (Waters). LC-MS data was acquired using MassLynx 4.2 (Waters). An ACQUITY UPLC BEH C18 column (1.7 μm, 2.1 mm × 100 mm) heated at 55 °C was utilised for chromatographic separation. The mobile phases were as follows; mobile phase A was acetonitrile:water (v/v 60:40) with 0.1% formic acid, mobile phase B was isopropanol:acetonitrile (v/v, 90:10) with 0.1% formic acid. An injection volume of 5 μL was used. The flow rate was set at 0.6 mL/min and the gradient elution began at 40% B and increased to 50% B over 30 s, then to 69% B at 1.8 min, with a final ramp to 88% B at 6 min. The gradient was reduced back to 40% B and held for 1 min to equilibrate column. Full MS spectra were obtained for the mass range m/z 50–2000, whilst infusing Leucine-Enkephalin (m/z 556.2766) as an online mass calibrant to retain mass accuracy. MS settings were as follows: Synapt G2-Si MS was operated in Q-ToF mode. Capillary voltage was set to 3.0 kV, sampling cone voltage was set to 40 V, source temperature was kept at 120 °C, desolvation temperature was set to 550 °C and desolvation gas flow was 900 L/h. MS$^E$ acquisitions used identical LC and MS conditions, with an added high energy ramp from 19 to 45 V.

**Sample sequence and quality control**. Pooled QC samples were used to check analytical reproducibility both during analysis and during the data processing stages[52]. QC samples were injected at the beginning of each analytical batch ($n = 3$), every 5th injection, and at the end of each analytical batch ($n = 2$). Samples from 274 participants were stratified and randomised into 15 equal analytical batches. Each batch was reconstituted on the day of analysis to maintain sample integrity. LC-MS$^E$ data were acquired for five sequential injections of a single pooled QC sample using an LC-MS$^E$ method in which all sampling preparation/handling, LC and MS conditions were identical to patient samples, except with an added high energy MS ramp.

**Data pre-processing and deconvolution**. LC-MS raw data were deconvolved using Progenesis QI (Waters, Wilmslow, UK). Peak picking, alignment, and area normalisation were carried out with reference to a pooled QC. The resulting peak table had 8765 metabolite features. Features that were absent in more than 10% of pooled QC injections throughout analysis were removed. From the remaining features those with more than 20% relative standard deviation (RSD) in peak intensity across pooled QC injections were also removed. The remaining peak set of 6202 metabolite features were robust features detected reproducibly throughout analysis within QC samples. The data were mean centred and auto-scaled and missing values were replaced with cubic spline interpolation in MATLAB 2019a (MathWorks) prior to statistical analysis.

LC-MS$^E$ raw data were deconvolved using Progenesis QI (Waters, Wilmslow, UK). Peak picking, alignment, and area normalisation were carried out using one of the QC data files as the reference. Significant features extracted from raw data were aligned to significant features in clinical samples, using a RT window ±15 s and mass tolerance ±10 ppm filters. Features were annotated using accurate mass match and tandem MS data with Lipid Maps, Lipid Blast, and METLIN. Mass tolerances of 10 and 30 ppm were applied for precursor and fragment ions, respectively. Compounds with a fragmentation score <20 were not annotated. Progenesis QI score, fragmentation score, and isotope similarity are reported for all annotations based on a combination of accurate mass and fragmentation data, see Supplementary Tables 4–6.

**Statistical analysis**. PLS-DA was performed for classification and prediction of data; resampling with replacement (bootstrapping) was used for model validation where the correct classification rates (CCRs) from the Y-variable were computed for the ($n = 250$) test data sets only. An in-house script was used in MATLAB (2019a) to perform PLS-DA. Univariate ROC analysis was performed in Origin (Version 2017, OriginLab Corporation, Northampton, MA, USA) and multivariate ROC curve-based exploratory analysis was executed using MetaboAnalyst Biomarker Analysis (Version 4.0) in which the data matrix was auto-scaled and PLS-DA was used for the classification method, and feature ranking method with a two latent variable input.

**Pathway analysis**. Mummichog analysis was performed using MetaboAnalyst (Version 4.0). During mummichog analysis a list of all m/z features ($L_{ref}$) and a refined list of significant m/z features ($L_{sig}$) were generated using Student's t-test as the discriminatory test (p-value < 0.05). Significant m/z features were mapped onto a combination of metabolic models: Kyoto Encyclopaedia of Genes and Genomes (KEGG), Biochemical Genetic and Genomic knowledgebase (BiGG) and the Edinburgh Model. Feature hits on known metabolite networks were tested against a null distribution produced from permutations of random m/z features from $L_{ref}$ to yield significance values of metabolites enriched within any given network[34].

**Reporting summary**. Further information on experimental design is available in the Nature Research Reporting Summary linked to this paper.

## Data availability

Raw and processed data sets generated during and/or analysed during the current study are available from MetaboLights Repository, https://www.ebi.ac.uk/metabolights/MTBLS2266 Study Identifier MTBLS2266. Annotation of metabolites utilised publicly available databases such as LipidMaps, METLIN and LipidBlast and HMDB. Source data are provided with this paper.

## Code availability

The code generated during the study are available from the corresponding author on reasonable request. Code to perform PLS-DA classification modelling is available at https://github.com/Biospec/cluster-toolbox-v2.0/blob/master/cluster_toolbox/plsda_boots.m

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

## Acknowledgements

We thank Michael J Fox Foundation (grant ref:12921) and Parkinson's UK (grant ref: K-1504) for funding this study. This work was supported by an EPSRC DTA grant to the School of Chemistry, which has funded the PhD project of E.S., a BBSRC DTP grant which has funded the PhD project of C.W.D. (BB/R505869/1) and the BBSRC (award BB/L015048/1) for instrumentation used in this work. We also thank our recruitment centres (See supporting information for lead personnel) for their enthusiasm and rigor during the recruitment process. We are very grateful to all the participants who took part in this study as well as PIs and nurses across all the recruiting centres. We also thank Richard Weller for feedback and discussions on sebum and dermatology.

## Author contributions

E.S.: Conception and design of work, data collection, data analysis and interpretation, drafting, and editing of article. D.K.T.: Conception and design of work, data analysis and interpretation and editing of article. D.S. and C.W.D.: Conception and design of work and data collection. R.B.: Conception of work and sample collection. J.M., M.S., T.K., A. M.R., and R.G.: Conception and design of work, interpretation of results and editing of article. P.E.B: Conception and design of work, supervision of data analysis, interpretation of results, and editing of article.

## Competing interests

The authors declare no competing interests.
