## [Peer Review File · Nature Communications]

Reviewers' comments:

Reviewer #1 (Remarks to the Author):

This is a potentially interesting manuscript which raises extremely interesting possibilities for biomarker assessment of Alzheimer's disease. As a reviewer I am however concerned by the methodology adopted to analyse lipids in sebum. The authors have adopted an LC-MS untargeted approach but have annotated peaks solely on the basis of database searching. This is an extremely inaccurate methodology and thus I would question the veracity of the data presented in Table 2 and Figure 3 and this question therefore undermines the results presented in Figure 4. Examination of the methodology presented in this paper describes the use of HPLC-MS with a C-18 reverse phase column. Whilst column chromatography will reduce the complexity of the m/z trace, thereby lowering the likelihood of ion suppression, the absence of the use of appropriate lipid class standards raises doubts about the assignation of molecular identities. Entering the m/z values quoted in Table 2 (666.6370, 638.6067, 610.5763, 825.6939, 764.5681, 414.4308, 358.3677, 194.1396, 550.6277, 368.4242) into the COMP_DB search on the LIPID MAPS website indeed reports the molecules quoted in Table 2 but it also suggest multiple alternative identities and in the absence of co-elution with appropriate standards the assignation cannot be accepted. Consequently the potentially important identification of biomarkers and altered pathways in the sebum of patients with Alzheimer's is not valid. If the authors can confirm the identities this would be. novel and important paper.

Reviewer #2 (Remarks to the Author):

The paper by Sinclair et al. holds a very interesting premise and implications for understanding factors that indicate systemic changes in brain degeneration disorders such as Parkinson's disease.

Major Concerns:

1) The paper is not yet clear in what the main contribution of the study would be. Unfortunately, the delineation of medicated- versus non-medicated PD is confusing. In the initial PLS-DA analysis, the authors state (line 176 and onwards) that:

VIP score examination of drug naïve PD vs. control and medicated PD vs. control models confirms that 10 variables (VIP > 1) are common between the two PD groups. To investigate biomarkers associated with the diagnosis of PD rather than disease stage stratification and to avoid possible effect of medication, the common metabolites between drug naïve and medicated PD analyses were evaluated further.

Then, in figures 3 and 4, analyses of drug naïve- and medicated PD cohorts are shown separately. It is not immediately clear to this reviewer why a selection of common features in the two cohorts was necessary only to then delineate these same measurements over the two categories. As 26 variables were identified in medicated PD, and 15 in drug naïve patients, it seems strange to exclude so many measurements if the two groups are compared post hoc.

2) Additionally, it is not ideal to have such variable ages comparing the healthy subject- and PD cohorts (55 years vs. 70 years, respectively). The authors do account for this potential bias by including age as a variable in the analysis, which ranked it as the least contributing variable in the analysis – this seems odd, as several papers have shown altered lipid metabolism in both aging and Parkinson disease (Rocha et al., 2015; Hallett et al, 2018; Huebecker et al., 2019).

3) Finally, the link between these metabolites and mitochondrial function is not immediately clear. The title reads 'a window into dysregulation of mitochondrial metabolism in Parkinson's disease', but ceramides, triacylglycerol and glycosphingolipid can all be metabolized in lysosomes. Relatedly, the authors write (line 258 and onwards) that:

'alteration to the expression of lipids within the sphingolipid pathway leads to lysosomal and mitochondrial degradation which are often implicated in the pathogenesis of neurodegenerative

diseases such as PD and Gaucher's disease'

which is a confusing statement, as the involvement of sphingolipid perturbations in these diseases is primarily associated with loss-of-function in the lysosomal enzyme glucocerebrosidase. Overall, the direct link to mitochondrial integrity in this study is not clear to this reviewer, and not necessary for the overall impact of the findings themselves.

References to more complete theoretical reviews on the current status of lysosomal, glycolipid, sphingolipid, lipid and inflammatory interactions on a systemic level leading to Parkinson's disease could be helpful to the readers, for example as described in Hallett et al 2019 [PMID: 31331333] or Isacson et al 2019 [PMID: 31649605].

Minor comments:

- Figure 3: the authors state that the chemical structure of each metabolite is shown underneath each box plot, but they cannot be found
- Table 2: the by far most differentially regulated metabolite (85% down in PD) was unassigned (no significant hit). Why did this metabolite avoid detection?
- Figure 4: the enrichment factor needs to be written on the figure - it's difficult to eyeball the size of the circles relative to the legend. The p-values of enrichment analyses on the two cohorts are also very different overall, why? Additionally, the steroid hormone biosynthesis pathway is only detectable in medicated subjects - where it is the most significant estimate. Does the medication modulate this pathway somehow?

Reviewer #3 (Remarks to the Author):

The manuscript submitted by Sinclair et al. describes the correlation between the sebum metabolome and Parkinson Disease (PD) with the aim of identifying potential new biomarkers. The topic should be of interest to a wide community entangled in PD research. Furthermore, the topic is timely and the approach of using sebum for determination of prognostic/diagnostic PD biomarkers is new and highly innovative. The manuscript is well written and conveys a clear message with not much to add. While the statistical part is well performed the analysis part of the manuscript could need some improvement. First of all, the annotation of lipids should rather be performed according to the official LIPID MAPS nomenclature (Liebisch et al., JLR, 2013) than according to the rather vague guidelines cited in the manuscript. Additionally, Table S4 shows some inconsistencies in annotations. While some m/z values like 816.7202 obviously refer to the neutral non-ionized lipid others like 553.4300 refer to molecular adduct ions. For the sake of clarity, please choose just one way of annotation. Furthermore it is questionable where ammonia adducts should come from when there was no ammonia in the mobile phase in first place. Additionally, ammonia adducts of PC (e.g. m/z 553.43) are highly unlikely, even with ammonia in the mobile phase. The same is true for H₂O loss at PC (m/z 792.5955), which would require either a free hydroxy group or complicated molecular rearrangements to say the least. The fatty alcohol reported at m/z 283.2885 should either be reported at m/z 266 (neutral mass) or m/z 284 (ammonia adduct)! Another issue are biologically rather rare lipids like the proposed ceramide C₄₁H₈₃N₁O₅ with an odd fatty acyl or sphingosine carbon number and two additional oxygens in comparison to 'regular' ceramides. When looking up this elemental composition in PubChem the vast majority of the 35 hits are not ceramides, which in turn means that this elemental composition could be associated with many other structures. For these reasons it would be good analytical practice to further corroborate the proposed lipid structures by MS/MS data.

We thank all of the reviewers for their constructive feedback on our manuscript and note their general support of this work. We have taken their comments and insights into consideration and amended the manuscript accordingly. Most crucially, *and despite not being in the lab, we have been able to analyse MS/MS fragmentation data which was taken alongside the LC-MS metabolomics data, to increase confidence in the annotations provided in the manuscript as requested.* We have addressed this and all of the other reviewer's concerns below.

Reviewer #1 (Remarks to the Author):

This is a potentially interesting manuscript which raises extremely interesting possibilities for biomarker assessment of Alzheimer's disease. As a reviewer I am however concerned by the methodology adopted to analyse lipids in sebum. The authors have adopted an LC-MS untargeted approach but have annotated peaks solely on the basis of database searching. This is an extremely inaccurate methodology and thus I would question the veracity of the data presented in Table 2 and Figure 3 and this question therefore undermines the results presented in Figure 4.

Examination of the methodology presented in this paper describes the use of HPLC-MS with a C-18 reverse phase column. Whilst column chromatography will reduce the complexity of the m/z trace, thereby lowering the likelihood of ion suppression, the absence of the use of appropriate lipid class standards raises doubts about the assignment of molecular identities. Entering the m/z values quoted in Table 2 (666.6370, 638.6067, 610.5763, 825.6939, 764.5681, 414.4308, 358.3677, 194.1396, 550.6277, 368.4242) into the COMP_DB search on the LIPID MAPS website indeed reports the molecules quoted in Table 2 but it also suggest multiple alternative identities and in the absence of co-elution with appropriate standards the assignment cannot be accepted. Consequently the potentially important identification of biomarkers and altered pathways in the sebum of patients with Alzheimer's is not valid. If the authors can confirm the identities this would be. novel and important paper.

As the reviewer appreciates in their comment, this is novel work in the field of Parkinson's disease and further in the analysis of sebum. Performing untargeted metabolomics is not novel but it's application using sebum for diagnostic insights to PD is. We agree that having definitive identification (Metabolomics Standards Initiative (MSI) Level 1 annotation) of each and every 'biomarker' peak would be ideal if the aim was to target a biomarker species. However, in this data driven untargeted metabolomics approach **the principle aim is to discover underlying differences between the two phenotypes.**

Annotation of identified features was perhaps unclear although we have adhered to Metabolomics Society's MSI. As the reviewer has rightly pointed, accurate mass match would generate multiple identifications and without running standards one cannot ascertain which of the multiple species it is. For an experiment that generates over 200 'significant biomarker peaks' for which accurate mass match may be an average of 5, an experimenter will be required to purchase 1000 standards for identification. This is a significant bottleneck in metabolomics and to overcome this burden for a data driven study like ours, MSI suggests identifying these annotations are putative identifications of level

3.¹ We had included this and clarify that we are not suggesting that our annotations are definite annotations or identifications of compounds but merely of the best accurate mass match. Further, the network analysis we perform, does not rely on how the metabolites are annotated, but how the accurate masses are measured.

To address the issues relating to biomarker identifications, we have now provided MS/MS fragmentation data to heighten confidence in our putative annotations for a number of our VIP compounds (VIP is from the PLS analyses) both in the main paper (Table 2A, *page 10*) and in the supporting information (Tables S4 and S5, *SI pages 9-13*). For a few of these significant species, predominantly the lower mass features in Table 2B (*page 11*) we were unable to obtain these MS/MS data (this is based on existing data as we have now been out of our lab for 12 weeks). The annotations supported by fragmentation data have been separated from those without the corresponding MS/MS validation. A list of all potential database matches (using Lipid Maps and METLIN) is listed for these compounds based on accurate mass (within 10 ppm). We have now removed the putative ID headings above the box plots and therefore nothing in this figure hinges on the veracity of the putative annotations presented in Table 2, they are merely associated with an accurately measured m/z value.

Furthermore, data presented in Figure 4 also do not rely on the annotations of the VIP features in Tables 2/S3/S4. The method utilised in the pathway analysis was Mummichog² network analysis, which uses correlations between accurate mass measurements that are mapped onto known human metabolic pathways. This methodology has been used in numerous metabolomics workflows.³⁻⁶ The original journal article describes the method as follows:

*“a novel approach to predict biological activity directly from mass spectrometry data without *a priori* identification of metabolites. By unifying network analysis and metabolite prediction under the same computational framework, the organization of metabolic networks and pathways helps resolve the ambiguity in metabolite prediction to a large extent.”¹*

We have summarised this workflow in our manuscript for the benefit of readers [page 13, lines 251-255]

“A prerequisite for traditional pathway analysis methods is the annotation of all analytically detected features via spectral and compound database matching. This is a major bottleneck in untargeted metabolomics workflows and due to the large number of features detected in this study, Mummichog analysis was employed”

And further, in the methods section [page 19, lines 420-428]:

*“Mummichog analysis was performed using MetaboAnalyst (Version 4.0). During mummichog analysis a list of all m/z features (L_{ref}) and a refined list of significant m/z features (L_{sig}) were generated using Student’s *t*-test as the discriminatory test (p -value < 0.05). Significant m/z features were mapped onto a combination of metabolic models: Kyoto Encyclopedia of Genes and Genomes (KEGG), Biochemical Genetic and Genomic*

knowledgebase (BiGG) and the Edinburgh Model. Feature hits on known metabolite networks were tested against a null distribution produced from permutations of random m/z features from L_{ref} to yield significance values of metabolites enriched within any given network.²

Reviewer #2 (Remarks to the Author):

The paper by Sinclair et al. holds a very interesting premise and implications for understanding factors that indicate systemic changes in brain degeneration disorders such as Parkinson's disease.

We thank the reviewer for their detailed evaluation of the manuscript. We have further explained our methods and rationale to address the major concerns and have revised some of the content in the manuscript, as follows.

Major Concerns:

1) The paper is not yet clear in what the main contribution of the study would be. Unfortunately, the delineation of medicated- versus non-medicated PD is confusing. In the initial PLS-DA analysis, the authors state (line 176 and onwards) that:

VIP score examination of drug naïve PD vs. control and medicated PD vs. control models confirms that 10 variables ($VIP > 1$) are common between the two PD groups. To investigate biomarkers associated with the diagnosis of PD rather than disease stage stratification and to avoid possible effect of medication, the common metabolites between drug naïve and medicated PD analyses were evaluated further.

Then, in figures 3 and 4, analyses of drug naïve- and medicated PD cohorts are shown separately. It is not immediately clear to this reviewer why a selection of common features in the two cohorts was necessary only to then delineate these same measurements over the two categories. As 26 variables were identified in medicated PD, and 15 in drug naïve patients, it seems strange to exclude so many measurements if the two groups are compared post hoc.

The two PD cohorts are shown separately throughout the paper from classification modelling and feature selection to pathway enrichment analysis. That being said, we focus on features selected from each independent analysis that are common between them. We use these features because they are discriminatory in two independent sets of PD patients and therefore we can establish that they are not an effect of disease stage or medication etc. which is an important property of a biomarker.

We do not exclude the non-common VIP features in each analysis, however, without further clinical information, which is not part of this preliminary study we do not comment on these features further. Figures 1 to 4 all show drug naïve and medicated PD cohorts separately and there is no mention of combining the cohorts in any part.

2) Additionally, it is not ideal to have such variable ages comparing the healthy subject- and PD cohorts (55 years vs. 70 years, respectively). The authors do account for this potential bias by including age as a variable in the analysis, which ranked it as the least contributing variable in the analysis – this seems odd, as several papers have shown altered lipid metabolism in both aging and Parkinson disease (Rocha et al., 2015; Hallett et al, 2018; Huebecker et al., 2019).

We agree with the reviewer for highlighting the effect of age on metabolism in various biofluids. It is one of the major confounding factors in many metabolomics studies. However, sebum is not a commonly studied biofluid so we would hypothesise that whilst altered lipid metabolism correlating to age has been established in alternative biological matrices, it does not translate directly to metabolites in sebum. We are measuring lipids as a secretion output on the skin and hypothesis that these may not mirror the same dysregulation as a function of age that has been reported for other biofluids. Our data show that measured lipids that are differential in classification of PD are in fact **not** influenced by age. This is detailed in the manuscript [page 9, lines 200-208]:

“To exclude the possible contribution of age to disease classification, age was included as a variable for further PLS-DA models, giving it equal weighting as any other measured variable. If age had any significance it had equal chances to contribute to the model and would be ranked as high as other measured variables. The difference in CCR between models with and without the inclusion of age were negligible (< 0.5 %), and VIP scores for the age variable were 1.17×10^{-11} and 2.11×10^{-11} for drug naïve and medicated PD models, respectively. In perspective, the variables were ranked at 6492 and 6498 out of a possible 6505 ranks, which strongly indicates that age is not a contributing factor for the separation presented.”

3) Finally, the link between these metabolites and mitochondrial function is not immediately clear. The title reads ‘a window into dysregulation of mitochondrial metabolism in Parkinson’s disease’, but ceramides, triacylglycerol and glycosphingolipid can all be metabolized in lysosomes. Relatedly, the authors write (line 258 and onwards) that:

‘alteration to the expression of lipids within the sphingolipid pathway leads to lysosomal and mitochondrial degradation which are often implicated in the pathogenesis of neurodegenerative diseases such as PD and Gaucher’s disease’

which is a confusing statement, as the involvement of sphingolipid perturbations in these diseases is primarily associated with loss-of-function in the lysosomal enzyme glucocerebrosidase. Overall, the direct link to mitochondrial integrity in this study is not clear to this reviewer, and not necessary for the overall impact of the findings themselves. References to more complete theoretical reviews on the current status of lysosomal, glycolipid, sphingolipid, lipid and inflammatory interactions on a systemic level leading to Parkinson’s disease could be helpful to the readers, for example as described in Hallett et al 2019 [PMID: 31331333] or Isacson et al 2019 [PMID: 31649605].

We thank the reviewer for alerting us to these two papers. We have taken the reviewer's comment into consideration and made the following changes to the manuscript to address them.

The two references suggested by the reviewer have been added to the main paper (*page 15, line 296*). We have altered the above referenced text to (*page 14-15, lines 293-296*):

"Perturbations within the sphingolipid pathway have been previously linked to defects in both lysosomal and mitochondrial metabolism with are often implicated in the pathogenesis of neurodegenerative disorders such as PD and Gaucher's disease"

We have also amended manuscript title to reflect the broader change The 'dysregulation of mitochondrial metabolism in PD' is no longer the title of the paper and we now replace mitochondrial metabolism for a broader topic of 'lipid metabolism' as the focus of the manuscript.

Minor comments:

- Figure 3: the authors state that the chemical structure of each metabolite is shown underneath each box plot, but they cannot be found

This has been changed.

- Table 2: the by far most differentially regulated metabolite (85% down in PD) was unassigned (no significant hit). Why did this metabolite avoid detection?

It did not avoid detection, but rather any identification, unfortunately, our searches of common databases using accurate mass and fragmentation MS/MS data did not yield any hits within the narrow mass tolerance applied (± 10 ppm). We keep the accurate mass tolerance low and conject this ion may be a conjugated or adducted form that cannot be matched to current reported annotations in databases.

- Figure 4: the enrichment factor needs to be written on the figure - it's difficult to eyeball the size of the circles relative to the legend.

The p-values of enrichment analyses on the two cohorts are also very different overall, why?

Additionally, the steroid hormone biosynthesis pathway is only detectable in medicated subjects – where it is the most significant estimate. Does the medication modulate this pathway somehow?

From the data we report, it could be speculated that steroid hormone biosynthesis may be due to medications. However, it should be noted that most of the Parkinson's participants have different dosage of medication that is prescribed to them based on their symptoms and severity of these symptoms. This is certainly one of the key areas that need more research and as we focus our future work for staging and grading of Parkinson's, a

dedicated study into targeted investigation of steroid hormone biosynthesis and other pathways is indeed planned.

Enrichment factors can be found for each pathway in the supplementary information for both drug naïve and medicated PD analyses. The figure highlights changes to enrichment between the two cohorts which is why only the common pathways are listed, we believe the differences are conveyed by the circle size, as depicted. Changes to the significance of metabolic pathways in line with disease progression is not unexpected, however, without further clinical input we cannot comment at this stage. Future studies will include clinical staging information and at that point we may be able to hypothesise why certain pathways become more or less significant as the disease progresses.

Reviewer #3 (Remarks to the Author):

The manuscript submitted by Sinclair et al. describes the correlation between the sebum metabolome and Parkinson Disease (PD) with the aim of identifying potential new biomarkers. The topic should be of interest to a wide community entangled in PD research. Furthermore, the topic is timely and the approach of using sebum for determination of prognostic/diagnostic PD biomarkers is new and highly innovative. The manuscript is well written and conveys a clear message with not much to add. While the statistical part is well performed the analysis part of the manuscript could need some improvement.

We thank the reviewer their useful comments on the manuscript, we have worked to address these as follows.

First of all, the annotation of lipids should rather be performed according to the official LIPID MAPS nomenclature (Liebisch et al., JLR, 2013) than according to the rather vague guidelines cited in the manuscript. Additionally, Table S4 shows some inconsistencies in annotations. While some m/z values like 816.7202 obviously refer to the neutral non-ionized lipid others like 553.4300 refer to molecular adduct ions. For the sake of clarity, please choose just one way of annotation.

The Lipid Maps nomenclature has been adopted as suggested. Measured m/z and neutral mass (where applicable) have been reported for all compounds and the annotation method is now consistent throughout (see Tables 2, S3, S4, S5).

Furthermore it is questionable where ammonia adducts should come from when there was no ammonia in the mobile phase in first place. Additionally, ammonia adducts of PC (e.g. m/z 553.43) are highly unlikely, even with ammonia in the mobile phase.

Although we did not add a source of ammonia to the mobile phase, we did use 10 mM ammonium formate in both mobile phases during method development immediately prior to analysing these samples and it is routinely used in this LC-MS instrument. Additionally, ammonia is present in all body fluids in the form of NH_4^+ , in particular it is excreted on the skin through sweat glands and it has been shown an intermediary metabolism product on the skin.⁷⁻⁹ We hypothesise that the skin may provide the source of ammonia we measure

as adductation in sebum analysis, particularly as our patient information leaflet specifies that the participant does not wash for at least 24 hrs prior to sebum collection.

The same is true for H₂O loss at PC (m/z 792.5955), which would require either a free hydroxy group or complicated molecular rearrangements to say the least. The fatty alcohol reported at m/z 283.2885 should either be reported at m/z 266 (neutral mass) or m/z 284 (ammonia adduct)! Another issue are biologically rather rare lipids like the proposed ceramide C₄₁H₈₃N₁O₅ with an odd fatty acyl or sphingosine carbon number and two additional oxygens in comparison to 'regular' ceramides. When looking up this elemental composition in PubChem the vast majority of the 35 hits are not ceramides, which in turn means that this elemental composition could be associated with many other structures. For these reasons it would be good analytical practice to further corroborate the proposed lipid structures by MS/MS data.

We have now reported compounds with putative annotations using tandem MS/MS data matched against Lipid Maps and Lipid Blast *via* Progenesis Q1 – as referred to above.

Features which could not be putatively annotated based on fragmentation data are listed in separate tables and all database matches using Lipid Maps and METLIN, within a narrow mass tolerance (± 10 ppm) have been listed. Whilst we agree there could be alternative IDs based on accurate mass searches against other database libraries, we have used Lipid Maps and METLIN because they best suit the nature of the biological matrix and are comprehensive.

References cited:

1. Sumner, L. W. *et al.* Proposed minimum reporting standards for chemical analysis Chemical Analysis Working Group (CAWG) Metabolomics Standards Initiative (MSI). *Metabolomics* **3**, 211–221 (2007).
2. Li, S. *et al.* Predicting Network Activity from High Throughput Metabolomics. *PLoS Comput. Biol.* **9**, (2013).
3. Rattray, N. J. W. *et al.* Metabolic dysregulation in vitamin E and carnitine shuttle energy mechanisms associate with human frailty. *Nat. Commun.* **10**, 1–12 (2019).
4. Khan, A. *et al.* Noninvasive Serum Metabolomic Profiling Reveals Elevated Kynurenine Pathway's Metabolites in Humans with Prostate Cancer. *J. Proteome Res.* **18**, 1532–1541 (2019).
5. Pamungkas, A. D., Medriano, C. A., Sim, E., Lee, S. & Park, Y. H. A pilot study identifying a potential plasma biomarker for determining EGFR mutations in exons 19 or 21 in lung cancer patients. *Mol. Med. Rep.* **15**, 4155–4161 (2017).
6. Johnson, C. H. *et al.* Metabolomics guided pathway analysis reveals link between cancer metastasis, cholesterol sulfate, and phospholipids. *Cancer Metab.* **5**, 1–9 (2017).
7. Downie, M. M. T. & Kealey, T. Human sebaceous glands engage in aerobic glycolysis and glutaminolysis. *Br. J. Dermatol.* **151**, 320–327 (2004).
8. Brusilow, S. W. & Gordes, E. H. Ammonia secretion in sweat. *Am. J. Physiol.* **214**, 513–517 (1968).
9. Czarnowski, D. & Gorski, J. Sweat ammonia excretion during submaximal cycling exercise. *J. Appl. Physiol.* **70**, 371–374 (1991).

REVIEWERS' COMMENTS

Reviewer #2 (Remarks to the Author):

The authors have responded in a satisfactory fashion to the issues raised by this reviewer in the original submission.

Refocusing the title (and emphasis) of the study to one of wide-spread alterations in lipid metabolism furthermore make the findings and implications easier to interpret in light of recent developments in the field of PD pathogenesis.

It is the opinion of this reviewer that the paper is now suitable for publication.

Reviewer #3 (Remarks to the Author):

All concerns are now addressed.